# Hurricane Arthur and its effect on the short-term variability of $pCO_2$ on the Scotian Shelf, NW Atlantic

Jonathan Lemay[1], Helmuth Thomas[1*], Susanne E. Craig[2], William J. Burt[1,3], Katja Fennel[1], and Blair J.W. Greenan[4]

[1]Department of Oceanography, Dalhousie University, Halifax, NS, Canada

[2]NASA Goddard Space Flight Center, Code 616, Greenbelt, Maryland 20771, USA

[3]now at: Dept. of Earth, Ocean and Atmospheric Sciences, University of British Columbia, Vancouver, BC, Canada

[4]Bedford Institute of Oceanography, Fisheries and Oceans Canada, Dartmouth, NS, Canada

[*]corresponding author: helmuth.thomas@posteo.org

Abstract

The understanding of seasonal variability of carbon cycling on the Scotian Shelf, NW Atlantic Ocean, has improved in recent years, however, very little information is available regarding its

short-term variability. In order to shed light on this aspect of carbon cycling on the Scotian Shelf we investigate the effects of Hurricane Arthur, which passed the region on July $5^{th}$ 2014. The hurricane caused a substantial decline in the surface water partial pressure of $CO_2$ ($pCO_2$), even though the Scotian Shelf possesses $CO_2$ rich deep waters. High temporal resolution data of moored autonomous instruments demonstrate that there is a distinct layer of relatively cold water

with low dissolved inorganic carbon (DIC) slightly above the thermocline, presumably due to a sustained population of phytoplankton. Strong storm-related wind mixing caused this cold intermediate layer with high phytoplankton biomass to be entrained into the surface mixed layer. At the surface, phytoplankton begin to grow more rapidly due to increased light.  The combination of growth and mixing of low DIC water led to a short-term reduction in partial

pressure of $CO_2$ until wind speeds relaxed and allowed for the restratification of the upper water column. These Hurricane-related processes caused a (net-) $CO_2$ uptake by the Scotian Shelf region that is comparable to the spring bloom, thus exerting a major impact on the annual $CO_2$ flux budget.



# 1. Introduction


Coastal oceans constitute the interface of four compartments of the Earth system: land, ocean, sediment, and atmosphere. Relatively shallow waters in the coastal oceans facilitate the immediate interaction between the atmosphere and sediment (e.g. Thomas et al. 2009, Thomas and Borges, 2012). Coastal oceans receive runoff from land (Chen and Borges, 2009) and are

impacted by the open oceans. They are a hot spot for biological production, accounting for a disproportionate amount of global ocean production relative to their surface area (Cai et al. 2003; Borges et al. 2005). Nutrients from rivers, the open ocean (e.g. Thomas et al., 2005), regenerated nutrients, and nutrients from shallow surface sediments fuel primary producers in coastal oceans. Consequently, coastal seas account for one-fifth to one-third of ocean primary production even

though they only account for 8% of the ocean surface area (Walsh, 1991). Due to their dynamic nature, coastal oceans experience much higher spatial and temporal variability (diel, seasonal, and annual) than the open oceans.

The Scotian Shelf is a coastal ocean and complex, multifactorial interactions result in challenges in determining the processes that control the high degree of variability reported in this

region (Shadwick et al. 2010, 2011, Signorini et al. 2013, Shadwick and Thomas 2014). Recent studies of the Scotian Shelf have focused primarily on monthly, seasonal, and inter-annual variability of carbon cycling (Shadwick et al. 2010, 2011, Shadwick and Thomas 2014, Craig et al. 2015), but these longer-term trends are overlain by significant short-term variability (e.g. Vandemark et al. 2011, Thomas et al. 2012) that to date, have remained relatively unstudied.

Storm activity on the Scotian Shelf has been shown to affect chlorophyll concentrations and

timing of the phytoplankton bloom (e.g. Fuentes-Yaco et al. 2005, Greenan et al. 2004), but little is known regarding the role of short-term variability in governing carbon cycling on the Scotian Shelf. A deepened mechanistic understanding is required to reliably assess the role of short-term events on longer-term variability and to facilitate future predictions with respect to climate change and ocean acidification. In the present study, we utilize autonomous moored sensors, and in-situ sampling to investigate the short-term variability of the $CO_2$ on the Scotian Shelf, with a focus on the impact of Hurricane Arthur, which passed the Scotian Shelf region on July 5[th], 2014.

## 2. Oceanographic Setting

The Scotian Shelf is located in the North West (NW) Atlantic Ocean at the boundary between the subtropical and subpolar gyres and extends from the Laurentian Channel to the Gulf of Maine spanning approximately the region of 43°N-46°N, 66°W-60°W (Fig. 1). The primary feature on the Scotian Shelf is the Nova Scotia Current, which is mostly derived from the Gulf of St. Lawrence (Dever et al. 2016).

The Scotian Shelf can be described as a 2-layer system in the winter (Fig. 2, 3a,b), when convective activity and wind-driven mixing control the mixed layer depth (MLD) and prevent stratification of the surface layer. During this period, the MLD is at its deepest and temperature and salinity are homogeneously distributed within the mixed layer. Any deeper layers are beyond direct impact of seasonal processes. As the MLD shoals during spring and summer due to lower wind speeds, warmer surface temperature, and fresh water input (Urrego-Blanco and Sheng 2012, Thomas et al. 2012), the Scotian Shelf transitions into a 3-layer system (Loder et al. 1997). The top layer of the 3-layer system in the summer is warm, shallow, and less saline as result of the increased discharge from the St. Lawrence (Loder et al. 1997). Below the warm, shallow

fresh layer is the cold intermediate layer (CIL), which consists of colder, saltier winter water. The third layer, beneath the CIL, consists of the warm slope waters from southern origin (e.g. Loder et al. (1997).

Sea surface temperature (SST) on the Scotian Shelf varies significantly over the course of the year, ranging from approximately 0°C during winter to a mean of 15°C during the summer, with peak highs of 20°C during the summer months (Fig. 2i). Surface salinity (Fig. 2ii) in the shelf region is relatively fresh, ranging from around 32 during the winter to 31.5 during late summer when the peak discharge of the St. Lawrence River arrives (Loder et al. 1997, Shadwick et al. 2011, Dever et al. 2016). Salinity increases further off the shelf as a result of the warm salty water from the Gulf Stream, which transports water south of the Scotian Shelf towards Western Europe.

Nitrate on the shelf is heavily influenced by the growth and decay of phytoplankton (Fig. 2). During the winter months, when phytoplankton productivity is low and wind-driven mixing of the water column is strongest, nitrate levels at the surface are high. As light levels increase in the spring, a phytoplankton population, dominated by diatoms, begins to grow, rapidly, depleting the nitrate reservoir in the surface waters (Craig et al., 2015). This short, but intense bloom heavily influences carbon cycling on the Scotian shelf. During this period, the region shifts from being a source of $CO_2$ to the atmosphere to a sink because of the biological $CO_2$ drawdown (Shadwick et al. 2010, 2011). Chlorophyll a concentration, the commonly used proxy for phytoplankton abundance (Fig. 2iv), demonstrate the intensity of the spring bloom during the months of March/April. The timing of the bloom varies between these two months depending on several factors including the onset of stratification and availability of light (Shadwick et al. 2010, Greenan et al. 2004, Ross et al. 2017). Once the phytoplankton bloom consumes the available

nitrate, the assemblage is taken over by smaller phytoplankton that prosper in the higher temperature, lower nutrient conditions (Craig et al. 2015; Li et al. 2006).

The subsurface chlorophyll maximum layer (SCML) is a feature almost ubiquitously found in stratified surface waters (Cullen 2015). During the late spring and summer period, the surface layer on the Scotian Shelf, which is nutrient poor following the intense growth of the spring bloom, becomes strongly thermally stratified. The phytoplankton, therefore, accumulate in deeper waters where nutrient concentrations are sufficient to support growth, but where there is still enough light available to drive photosynthesis (e.g. Cullen 2015). This occurs at the nutricline, i.e. the transition from the warm, nutrient poor surface layer to the cooler, comparatively nutrient rich second layer. Additionally, in these lower light conditions, phytoplankton often employ the survival strategy of photoacclimation, whereby they increase their intracellular chlorophyll concentration to maximize light absorption. This can result in an increased ratio of chlorophyll to carbon (Chl:C ratio) at the SCML (Cullen 2015). There is a suggestion of this summertime SCML in the climatological data from the region (Fig. 2iv) and in a recent glider study of the Scotian Shelf by Ross et al. (2017).

Observational studies reveal the Scotian Shelf to be a source of $CO_2$ to the atmosphere, except during the period of the spring bloom (Shadwick et al. 2010, 2011, see also Signorini et al. 2013 for discussion). Fluxes of $CO_2$ to the atmosphere are highly variable outside of the spring bloom period (Shadwick et al. 2010). Wind speeds impact the mixed layer depth, which in turn, can impact $CO_2$ fluxes on the shelf (Shadwick et al. 2010, Greenan et al. 2008). DIC increases with depth, which means mixing caused by strong wind events can bring carbon rich water to the surface. Shadwick et al. (2010, their Fig. 8) demonstrate how weather patterns can have a significant impact on monthly variation of $CO_2$ flux. The strength, timing and frequency

of winter storms impact the timing of the spring bloom (Shadwick et al. 2010). Spectral analyses

have shown that storm events occur at periods of 6 days and 3 weeks (Smith et al. 1978,

Shadwick et al. 2010, Thomas et al. 2012).

A significant contributor to annual storm activity on the Scotian Shelf comes from

hurricanes, with the 2003 hurricane season generating 14 named hurricanes in the Atlantic Ocean

(Fuentes-Yaco et al. 2005).  Hurricanes that affect the Western North Atlantic are formed mostly

in the Eastern Atlantic Ocean near Africa (Fuentes-Yaco et al. 2005).  After formation, the

hurricanes move westward on the trade winds, veer northeast around 30° to 35°N as they meet

the eastern prevailing winds from North America, and move towards, and often over, the Scotian

Shelf and/or the Newfoundland Shelf (Fuentes-Yaco et al. 2005). Hurricanes passing through the

northwest Atlantic can entrain cold nutrient rich water to the surface, which has been found to

stimulate primary production (Fuentes-Yaco 1997, Platt et al. 2005, Han et al. 2012).  The timing

of these storms has also been found to affect the timing and strength of the spring phytoplankton

bloom (Greenan et al. 2004).

In this paper, we will focus on the effect of the passage of Hurricane Arthur on $pCO_2$

observed at our study site on the Scotian Shelf. We will consider the partial pressure of $CO_2$,

$pCO_2$, conditions before, during, and after the storm's passage using highly temporally resolved

measurements, and present mechanistic explanations for the observed phenomena.

## 3. Methods

3.1 Sampling Procedures

The CARIOCA buoy used in this study was equipped with sensors to acquire hourly measurements at the surface (approximately 1m depth) for temperature, conductivity, the partial pressure of $CO_2$ (pCO$_2$), salinity, sea surface temperature (SST), and chlorophyll-a fluorescence between February $20^{th}$ to December $31^{st}$, 2014. An automated spectrophotometric technique was used to estimate pCO$_2$, and is fully described elsewhere (Bates et al. 2000; Bakker et al., 2001; Bates et al. 2001; Hood and Merlivat, 2001). Conductivity and temperature were measured using a SeaBird conductivity sensor (SBE 41) and a Betatherm thermistor respectively. A WETstar fluorometer (WETLabs) measured chlorophyll-a (Chl-a) fluorescence. The buoy was deployed at the Halifax Line Station 2 (HL2; 44.3N, 63.3W, ~30km offshore from Halifax, Nova Scotia) from February 2014 to January 2015 in addition to other deployments that took place between 2007 and 2012 (e.g., Thomas et al. 2012).

From April 2007 to the end of July 2007, a SeaHorse moored vertical profiler was placed at the location of HL2, where it acquired profiles from the surface to a depth of approximately 100 m every 2 hours. It was equipped with temperature, salinity, and Chl-a fluorescence sensors. A complete description of the SeaHorse operation and sensor suite can be found in Greenan et al. (2008), or Craig et al. (2015).

Water column samples were collected through the semi-annual Atlantic Zone Monitoring Program (AZMP) operated by the Canadian Department of Fisheries and Oceans. The AZMP cruises occur during the Spring (April) and Fall (September – October) every year. Bi-weekly sampling of HL2 is also conducted whenever weather permits. Water samples are collected using 10 L Niskin bottles mounted on a 24-bottle rosette with a SeaBird CTD. Collected samples are then poisoned with mercury chloride (HgCl$_2$) to prevent biological activity before the DIC concentration was measured using a VINDTA 3C system (Versatile Instrument for the

Determination of Titration Alkalinity by Marianda). This was also used to determine alkalinity

(TA) and DIC, and the measurement method is described in full detail by Johnson et al., 1993,

Fransson et al., 2001, or Bates et al., 2005. Certified reference material was provided by Prof. A.

Dickson (Scripps Institution of Oceanography) to determine the uncertainty of DIC and TA to ±2

and ±3 µmol kg$^{-1}$, respectively.

Non-photochemical quenching (NPQ) in phytoplankton is a mechanism by which excess

absorbed solar radiation can be dissipated in pathways other than Chl-a fluorescence (e.g., such

as heat), and can reduce chl-a fluorescence by up to 80% (Kiefer, 1973). In order to minimize the

impact of NPQ, only nighttime fluorescence from 0500UTC was used in analyses of Chl-a

fluorescence. Chl-a fluorescence was regressed against chl-a concentration determined from

fluorometric analysis of *in situ* water samples; this enabled creation of a calibration curve for

both the CARIOCA and SeaHorse data to allow comparison of measurements.

3.2 Computational Analysis

Temperature normalized pCO$_2$ was calculated using the equation from Takahashi et al.

(2002) (Equation1).

$$pCO_2\left(T^{mean}\right)=pCO_2^{obs}\left[\exp\left(0.0423\left(T^{mean}-T^{obs}\right)\right)\right] \tag{1}$$

This normalization removes the thermodynamic effects of temperature on pCO$_2$ and

reveals the non-temperature related, i.e., largely biological effects on pCO$_2$. The mean

temperature used for this calculation is 10°C.

Using the method developed by Friis et al. (2003), DIC is normalized to salinity to

remove the overlying salinity signal to better determine biological and anthropogenic impacts.

DIC$^S$ represents DIC normalized to salinity, S$^{obs}$ represents the measured salinity value, DIC$^{obs}$ represents the measured DIC value, S$^{ref}$ represents the salinity standard used to calculate DIC$^S$, which in this case is 32, and DIC$^{S=0}$ represents thefreshwater end member, which is 601 µmol DIC kg$^{-1}$ taken from Shadwick et al. (2011).

$$DIC^S = \frac{DIC^{obs} - DIC^{S=0}}{S^{obs}} * S^{ref} + DIC^{S=0} \qquad (2)$$

Sea-air fluxes from the CARIOCA dataset were calculated using the flux calculation functions from Wanninkhof (2014) (Equation 3).

$$F = -0.251 * U^2 * \left(\frac{Sc}{660}\right)^{-0.5} * K_0 * \left(pCO_2^{Obs} - pCO_2^{Atm}\right) \qquad (3)$$

Where F is in 10$^{-5}$mol m$^{-2}$ h$^{-1}$, U is wind speed (m s$^{-1}$), Sc is the Schmidt number, $K_0$ is gas solubility (mol L$^{-1}$ atm$^{-1}$), pCO$_2^{Obs}$ (µatm) is observed pCO$_2$, and pCO$_2^{Atm}$ (µatm) is atmospheric pCO$_2$, for which 400 µatm is used. The widely used flux calculations from Wanninkhof (2014) were used, and have an estimated 20% level of uncertainty. Full details regarding the flux equation can be found in Wanninkhof (2014).

3.3 Comparison CARIOCA / SeaHorse

For mechanistic analysis, we use the SeaHorse vertical profiler from 2007 to help underpin the observations made from the 2014 CARIOCA dataset. Chl-a concentration determined in the laboratory using fluorometric analysis of *in situ* water samples were regressed against factory calibrated nighttime fluorescence from the CARIOCA and SeaHorse data sets. r$^2$ (RMSE) values were found to be 0.532 (0.2 mg m$^{-3}$) and 0.743 (0.4 mg m$^{-3}$) for the CARIOCA

and SeaHorse, respectively. The poor agreement between the bottle and fluorescence Chl-a estimates is unsurprising since factory conversions of fluorescence to chlorophyll concentration rarely correspond well. This is due to several factors that include differences in fluorescence yield between the factory calibration standard and natural phytoplankton, differences in the water mass sampled (small volume illuminated by the fluorometers versus the larger water mass sampled by the Niskin bottle) and the fact that both estimates are subject to significant uncertainties. For these reasons, fluorescence estimates of Chl-a will be used in a qualitative manner to examine patterns and trends, rather than to determine exact concentrations.

A subset of the CARIOCA data collected from June 9th-17th (year days 160-168) 2007 were used to compare with the Seahorse data (Fig. 4). During this time, the CARIOCA and Seahorse instruments were simultaneously deployed, allowing for a comparison between the two datasets. The fluorescence for both data sets were converted to chlorophyll using the calibrations curves described above. Both sets showed chlorophyll of similar magnitude, as well as a similar trend over the time series (data not shown).

## 4. Results & Discussion

4.1 Observations of $pCO_2$, wind speed and fluorescence in 2014

Annual $pCO_2$ data from the 2014 CARIOCA dataset reveals that there is significant variability over the course of 2014 (Fig. 5a). Although impacted by the variability, the key annual features are obvious and include: the phytoplankton bloom (year days 80-110), a summer baseline (year days 150-300), and a winter baseline (year days 50-75 and 300-365). The variability (or amplitude of variability) in $pCO_2$ is more pronounced during the summer months compared to the winter and spring bloom periods (see also Thomas et al., 2012). The low

variability during the winter and the bloom is likely a result of the deeper, homogenous surface

mixed layer, which in turn acts as a buffer for any short-term variability. The data used in the

present paper reflect the reoccurring winter storm pattern, with a periodicity of approximately 6

days as reported earlier for the region (Smith et al. (1987), Shadwick et al. (2010), or Thomas et

al. (2012)).

Wind speeds for 2014 (Fig. 5b) show that during the winter, winds are stronger on the

Scotian Shelf, with higher storm frequency, while wind speeds are generally lower during the

spring and summer months. During the period of Hurricane Arthur, wind speeds of up to 30 m s$^{-1}$

were observed.

Fluorescence over the year (Fig. 5c) clearly shows the spring bloom increase of up to a

factor of 4 above the winter baseline. Similarly, during Hurricane Arthur, there is a doubling in

fluorescence above the summer baseline values compared to the adjacent days. Later in the year,

around year day 300, the fluorescence shows somewhat elevated values due to the minor fall

bloom that occurs as increased wind speeds begin to deepen the mixed layer bringing nutrients to

the surface (Greenan et al. 2004).

4.2 Hurricane Arthur

A prevalent feature of the time series is the sharp decrease in $pCO_2$ as wind speeds

increase during the hurricane (Fig. 5, days 186-193). Dissolved inorganic carbon increases with

depth in this region (Shadwick et al. 2014), therefore it is expected that increased wind speed

would increase $pCO_2$ as more carbon rich water is mixed to the surface.  However, wind and

$pCO_2$ are negatively correlated (r =-0.77, significance level, $\alpha = 0.05$; data not shown),  for the

whole year.  Decreases in $pCO_2$ with increases in wind are most evident from spring to early fall.

This coincides with the period where the water column becomes a 3-layer system as a result of solar insolation and increased discharge from the Gulf of St Lawrence (Loder et al. 1997). For this study, we chose the most prominent decrease in $pCO_2$, which occurred during Hurricane Arthur on July $5^{th}$. The underlying assumption is that Hurricane Arthur can be compared to other periods where low $pCO_2$ is correlated with high wind events within the spring to early fall period.

To identify the cause of the decrease in $pCO_2$ when wind speeds are high, surface water properties observed during the Hurricane Arthur period are examined in detail (Fig. 6). SST drops by roughly 6°C over a half day period indicating that water from the cold intermediate layer (CIL) was mixed into the surface layer by strong winds, causing rapid cooling (Fig. 6a). However, upon close inspection, it can be seen that the $pCO_2$(Tmean) decrease occurs prior to the temperature decrease, indicating that a non-temperature related $CO_2$ uptake process is at play before wind-driven CIL entrainment occurs (Fig. 6a). The disconnect between temperature and $pCO_2$ can be explained by considering the position of the subsurface chlorophyll maximum layer (SCML, Figure 3). This layer straddles the thermocline between layers 1 and 2, where phytoplankton can utilize nutrients diffusing across the boundary, but still receive enough light for photosynthesis. As the upper layer is mixed by wind, these phytoplankton are redistributed throughout the upper layer where they experience increased light exposure (compared with that at the SCML) that allows them to photosynthesize more efficiently and, therefore, draw down more $CO_2$.

Following the initial mixing of the surface layer, mixing energy then becomes sufficient to entrain waters from the deeper CIL between yeardays 186 and 187. This results in a rise in $pCO_2$(Tmean), a decrease in temperature (Fig. 6a) and increase in salinity (Fig. 6d). The increase

in salinity occurs in two separate steps (Fig. 6d, dashed grey box): The first coincides with the sharp decline in $pCO_2(T_{mean})$, indicating the redistribution of phytoplankton from the SCML throughout the surface layer and their corresponding uptake of $CO_2$. This is also evident in an initial increase in fluorescence prior to the salinity maximum (Figs. 6c, d). The second step aligns with the sharp increase in $pCO_2(T_{mean})$ pointing to continued vertical mixing into deeper saline waters rich in DIC from the CIL. When compared to the wind speeds (Fig. 6c), the second step also occurs during the wind speed maximum, when mixing would be at its strongest. Fig. 2(ii) shows the 3-layer system during the summer (dark blue, blue and yellow layers), with approximately one salinity unit difference between each of the climatological mean layer values. The magnitude of salinity change during Hurricane Arthur is comparable.

Chlorophyll-a fluorescence increases by approximately 40% during the hurricane, indicating that the mixed conditions of the water column favor phytoplankton growth (Fig. 6c). Nutrients at the surface are depleted during the summer months (Fig. 2iii) and, therefore, the response of the phytoplankton implies that the hurricane mixed nutrients upward from deeper in the water column. This line of argument is also supported by the observed corresponding salinity increase. Wind-driven mixing breaks through the fresh water layer at the surface, reaching into the deeper saline waters of the CIL where nitrate is more abundant. Interestingly, despite the increase in chlorophyll-a fluorescence (and implied phytoplankton abundance) during yeardays 187-190 (Fig. 6c) , $pCO_2(Tmean)$ continues to rise. This suggests that, despite increased primary productivity produced by the mixing event, the study site is a net source of $CO_2$ during this time as the entrainment of $CO_2$-rich deeper waters out-competes the effects of photosynthetic $CO_2$ fixation.

Normalizing the DIC observations from HL2 to a constant salinity (of 32) reveals the biological DIC fingerprint (Fig. 7b). This approach yields a minimum in DIC in the subsurface layer at a depth of approximately 20-25 m, which indicates DIC uptake by phytoplankton. Further support for the existence of this enclosed layer is provided by a study by Shadwick et al. (2011), in which negative apparent oxygen utilization (AOU) values at this depth level were observed during the summer period. The enclosed layer is both sufficiently shallow for photosynthesis to occur and sufficiently deep to supply the required nutrients through vertical diffusion across the nutricline (Fig. 2). When comparing the temperature minimum and salinity maximum from Figure 6 with the T/S profile of Figure 7 (see also below discussion of Fig. 9), they reveal a deepening of the mixed layer to around 50 m, which matches well with the DIC profiles. Figure 7a also shows that the density steadily increases with depth, and that the DIC minimum lies below the upper part of the mixed layer in a stable layer between waters of lower (above) and higher (below) density.

To shed light on processes occurring within the CIL, we employ high-resolution water column data at HL2 collected from the 2007 SeaHorse vertical profiler. Although the Seahorse data was collected during a different year, we assume that the observed features are present every year as characteristics of the overall system. The data from the SeaHorse profiler reveal a variable but persistent chlorophyll-a maximum below the surface post-spring into summer (Fig. 8). This persistent chlorophyll-a maximum occurs at roughly 25 m below the surface. This matches well with the profiles in Figure 7 that display normalized DIC minima at roughly the same depth.

A snapshot of SeaHorse surface profiles from June 9-17[th] 2007 was extracted and compared to data from the CARIOCA buoy during the same period (Fig. 4, see methods). As

with Hurricane Arthur (Fig. 6), a passing storm (of weaker strength) during this period shows the

same negative correlation between $pCO_2$ and wind speed. $pCO_2$ decreases for a period of time as

wind speeds increase.  There is also an increase in chlorophyll-a for both the CARIOCA and

SeaHorse data (Fig. 4), showing that both instruments detect the increase at a similar rate.

The selected 3 days shown in Figure 9 reveal the evolution of the water column during

the 2007 storm event (same event as Figure 4).  Before the storm there is a sub-surface

chlorophyll maximum, which is below the maximum of the density gradient.  The Brunt-Väisälä

frequency (Fig. 9) also shows stable stratification at roughly 18 m depth on June 9[th], followed by

stable stratification at 38 m for June 15[th], and on June 17[th], stratification stabilizes further up the

water column at 25m.  Once the storm approaches the water column becomes mixed, increasing

surface salinity and chlorophyll-a as well as homogenizing water density for the top 40 m.  In

this example however, temperature does not decrease at the surface as in Figure 6.  However,

Hurricane Arthur was a much stronger storm that resulted in deeper mixing of the water column

and more cooling of the SST.  When the storm subsides, the water column restores within 2 days

to its original state.  Surface chlorophyll and salinity return roughly to their pre-storm levels, and

the SCML is again below the density gradient. The data presented in Fig. 9 - in particular

temperature - show, lateral processes, either cross shelf or along shelf may have impacted the

system, as well. These features, however, cannot be further resolved referring to single-point

moored observations.

With water column data including fluorescence, Figure 6c is then analyzed to determine

how much of the fluorescence is attributed to new growth or mixing. As discussed by Cullen

(2015) the SCML contains a higher ratio of chlorophyll-a to carbon as a result of survival

strategy in reduced light, therefore it can be speculated that the rapid increase of fluorescence

could be the result of redistributed cells rather than new production. Integrating salinity over a depth of 50 m for June 9th and June 15th (Fig. 9) yields a constant salinity inventory of 1474 m and 1482 m, respectively [unit {m}: salinity{unitless} * integration depth{m}]. On the assumption that mixing is conservative, integration of chlorophyll-a for June 9th and June 15th is also performed. The results were 105 mgChl m$^{-2}$ and 158 mgChl m$^{-2}$ respectively, indicating that the majority of chlorophyll a (approximately 2/3) observed at the surface is redistributed over the 5 day period the integration took place.  The growth of the remaining 53 mgChl m$^{-2}$ (approximately 1/3) can be attributed to rapid phytoplankton growth that would be expected to take place as a result of nutrients being mixed to the lit surface layer. This helps explain the rapid increase in fluorescence observed in Figure 6 as most of the increase is due to redistributed phytoplankton from the SCML.

4.3 Impact of Hurricane Arthur on Carbon Cycling

In order to quantitatively estimate the direct impact Hurricane Arthur had on carbon cycling, air-sea fluxes were calculated for 2014 (Table 1).  The average daily flux for July was 0 mmolC m$^{-2}$ day$^{-1}$, however when the impact of Hurricane Arthur is removed from the average the new flux value is -7 mmolC m$^{-2}$ day$^{-1}$.  If Hurricane Arthur was averaged alone, the flux would be 19 mmolC m$^{-2}$ day$^{-1}$, nearly half the rate observed during the phytoplankton bloom (45 mmolC m$^{-2}$ day$^{-1}$). The impact of the hurricane was substantial enough to cancel out the overall emission of $CO_2$ to the atmosphere for the month of July.  This indicates that short-term storm events can have a significant impact on annual $pCO_2$ cycling for the Scotian Shelf in the regions affected by the storm.

# 5. Conclusions

The data provides compelling evidence that there is an interaction between wind speed, $pCO_2$, and sub-surface phytoplankton. However, the timing of a storm event dictates the strength of its impact. Previous work has shown that deeper water is rich in DIC compared to the surface, and it was expected that mixing of deeper water should increase $pCO_2$ as a result. However, sub-surface phytoplankton has a relatively strong influence on carbon cycling during storm events. The effects of storms on $pCO_2$ vary based on whether the water column is a 2- or 3-layer system, and their timing during these 2- and 3-layer periods. Hurricane Arthur was a special case in that it impacted the shelf while it was in its 3-layer phase. During this time, the entrained layer was stable as a result of the warm freshwater cap at the surface. This allowed phytoplankton to thrive at the boundary of the surface layer and the CIL, where there are more nutrients than in the upper mixed layer, but still enough light to drive photosynthesis. When the storm arrived and perturbed this enclosed layer, it caused a sharp decrease in $pCO_2$. It might appear evident that if there were changes in storm and wind characteristics, these would impact $CO_2$ fluxes on the Scotian Shelf. However so far, no clear trends for changes in such characteristics are evident (e.g. Brickman et al. 2013).

The study presented in this work largely rests on data from moored autonomous instruments such as the CARIOCA buoy, which supply observational data with high temporal coverage. The complementing use of SeaHorse data has expanded the observations into the vertical dimension, which facilitates the consideration of water column properties and their influence on the surface water $CO_2$ variability. In observational studies, a balance must always be found between temporal and spatial coverage, as for example discussed by Schiettecatte et al., (2007). It is clear that the use of data from moored instruments provides the high temporal

resolution data needed to understand high-frequency variability. This strength of this study is at the expense of spatial coverage, and accordingly, we cannot fully exclude lateral processes,

which might contribute to the variability of the $CO_2$ system as observed by our instruments.

*Acknowledgements:* We are grateful to BIO staff supporting mooring operations. This work was supported by MEOPAR and TOSST.

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

## Figure Captions

Fig. 1: Regional view of the Scotian Shelf with primary currents shown. The red star depicts the location of the Seahorse and Carioca moorings. Reprinted with permission from Shadwick et al. (2010) ©Shadwick et al. 2010

Fig. 2: Climatologies for the Scotian Shelf, observed at station HL2 (44.299°N 63.247°W). i) Temperature, ii) salinity, iii) nitrate, and iv) Chl-a. Reprinted with permission from Shadwick et al. (2011). © Elsevier

Fig. 3: Schematic demonstrating the evolution of the water column over the course of the year. The dashed line intersecting panels i), ii), and iii) represents the mixed layer. SCML in green represents the sub-surface chlorophyll maximum layer. Temperature and Salinity profiles provide an idealized view of the upper water column where panel a) corresponds to panel i), b) to ii), and c) to iii).

Fig. 4: 2007 CARIOCA data set for June 9th-17th from the with the x-axis representing year days. The black line is $pCO_2$ (µatm), red line is wind (m s-1), blue line solid is calibrated CARIOCA fluorescence (mg m$^{-3}$), and the blue dashed line is calibrated SeaHorse fluorescence (mg m$^{-3}$).

Fig. 5: 2014 time series data collected from the CARIOCA buoy with the x axis representing year day. Panel a shows $pCO_2$ in µatm, revealing a large amount of variation over the course of the year; with a minimum during the spring bloom and a high maximum over the course of the summer. Panel b shows wind speeds in m s$^{-1}$, with higher wind speeds during the winter period and lower speeds during the summer. Panel c shows fluorescence in arbitrary units with a spring broom clearly visible, and the rest of the year with generally low values. There is some evidence at a prolonged fall bloom after year day 300. The red bands represent the period where Hurricane Arthur (July 5th 2014) took place, and was selected as this feature stands out amongst the others.

Fig. 6: Observations during Hurricane Arthur taken from the CARIOCA 2014 dataset, with the x axis representing year days. Each panel has pCO2 in black (µatm) and pCO2(Tmean) (µatm) in blue; with a different variable in each panel overlain: (a) temperature, (b) wind speed, (c) flourescence, and (d) salinity. The grey box in panel d is used to highlight the change in salinity.

Fig. 7: Pre- (a) and post storm (b) vertical profiles taken from in-situ samples for HL2 collected in 2014. a) salinity, and temperature (°C), and density (kg m$^{-3}$) were collected on June 28th, 7 days before Hurricane Arthur. b) Post-storm DIC (µmol kg$^{-1}$) profiles for July 22 and August 3, 2014 collected at HL2 along with their corresponding $DIC_{norm}$ profiles normalized to a salinity value of 32, revealing the reestablishment of the pre-storm situation.

Fig. 8: SeaHorse vertical time series data collected at HL2. Fluorescence data is in mg m-3 and calibrated to in situ bottle data collected at HL2. White gap represents when the mooring was removed from the water for repairs. The black line represents the mixed layer depth in meters. The red line is the DIC$_{norm}$ profile (Aug. 3$^{rd}$, 2014) from Figure 7, with its scale at the top right of the figure. Please note that the DIC profile is collected from the 2014 year, while the SeaHorse
data is from 2007. This comparison is made to help underpin the mechanistic understanding of the water column structure.

     Fig. 9: SeaHorse snapshots of a storm event between June 9th to June 17th 2007. X-axis contains chlorophyll (mg m$^{-3}$), salinity, temperature (°C) and density (kg m$^{-3}$); and y-axis is depth in metres. Wind speeds (m s$^{-1}$) for each day (5:00am to correspond with time of SeaHorse data)
are included in each panel. Fluorescence values are calibrated to in-situ bottle data collected at HL2. The right hand side panels show the Brunt-Väisälä frequency for the respective days.

50

Table 1: Average daily sea-air fluxes (mmol m$^{-2}$day$^{-1}$) for each month available for the 2014 year using the Wanninkhof 2014 method. July is broken into 3 components: the month as a whole, the 8 days Hurricane Arthur impacted pCO$_2$, and the remaining 22 days averaged without hurricane Arthur. pCO2 (µatm), wind speed (m s$^{-1}$), temperature (°C), and salinity are averaged for each month (or segment in the case of Arthur and No Arthur).

625

| Month | CO$_2$ Flux (mmol m$^{-2}$ day$^{-1}$) | pCO$_2$ (µatm) | Wind Speed (m s$^{-1}$) | Temperature (°C) | Salinity |
|---|---|---|---|---|---|
| March | 18 | 374 | 14.9 | 0.1 | 30.9 |
| April | 45 | 316 | 14.5 | 1.2 | 31.2 |
| May | 2 | 395 | 9.0 | 4.5 | 31.2 |
| June | -3 | 430 | 9.1 | 9.9 | 30.9 |
| July | 0 | 423 | 12.2 | 12.6 | 31.1 |
| Arthur | 19 | 385 | 14.9 | 9.7 | 31.2 |
| No Arthur | -7 | 436 | 11.2 | 13.6 | 31.0 |
| August | -27 | 506 | 10.8 | 18.7 | 30.8 |
| September | -30 | 481 | 13.4 | 17.8 | 30.8 |
| October | -5 | 409 | 17.3 | 15.0 | 30.9 |
| November | 4 | 405 | 17.5 | 10.3 | 30.5 |
| December | -8 | 413 | 16.3 | 5.5 | 30.5 |

630

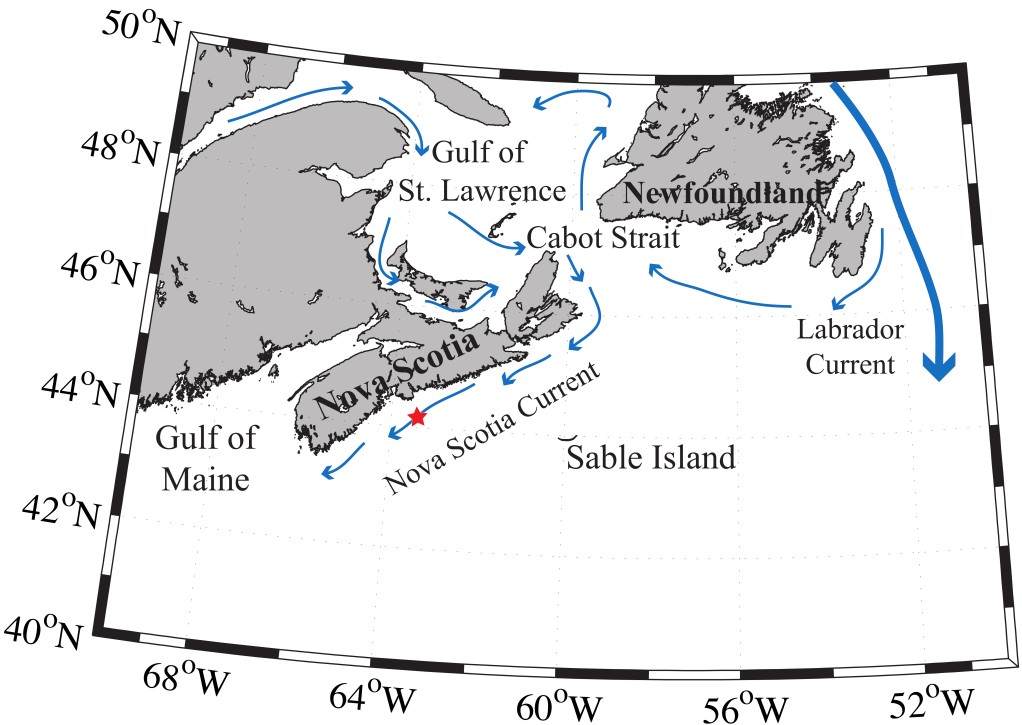

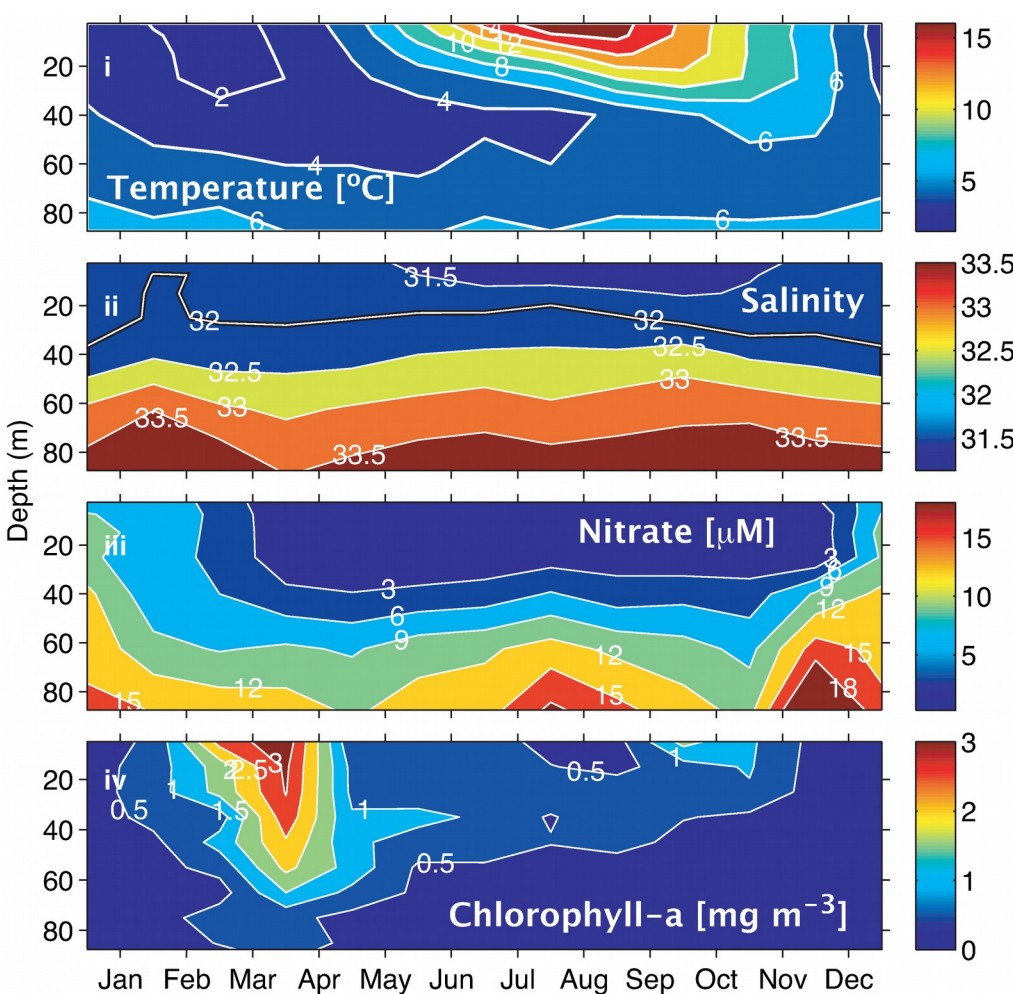

Lemay et al., Figure 2

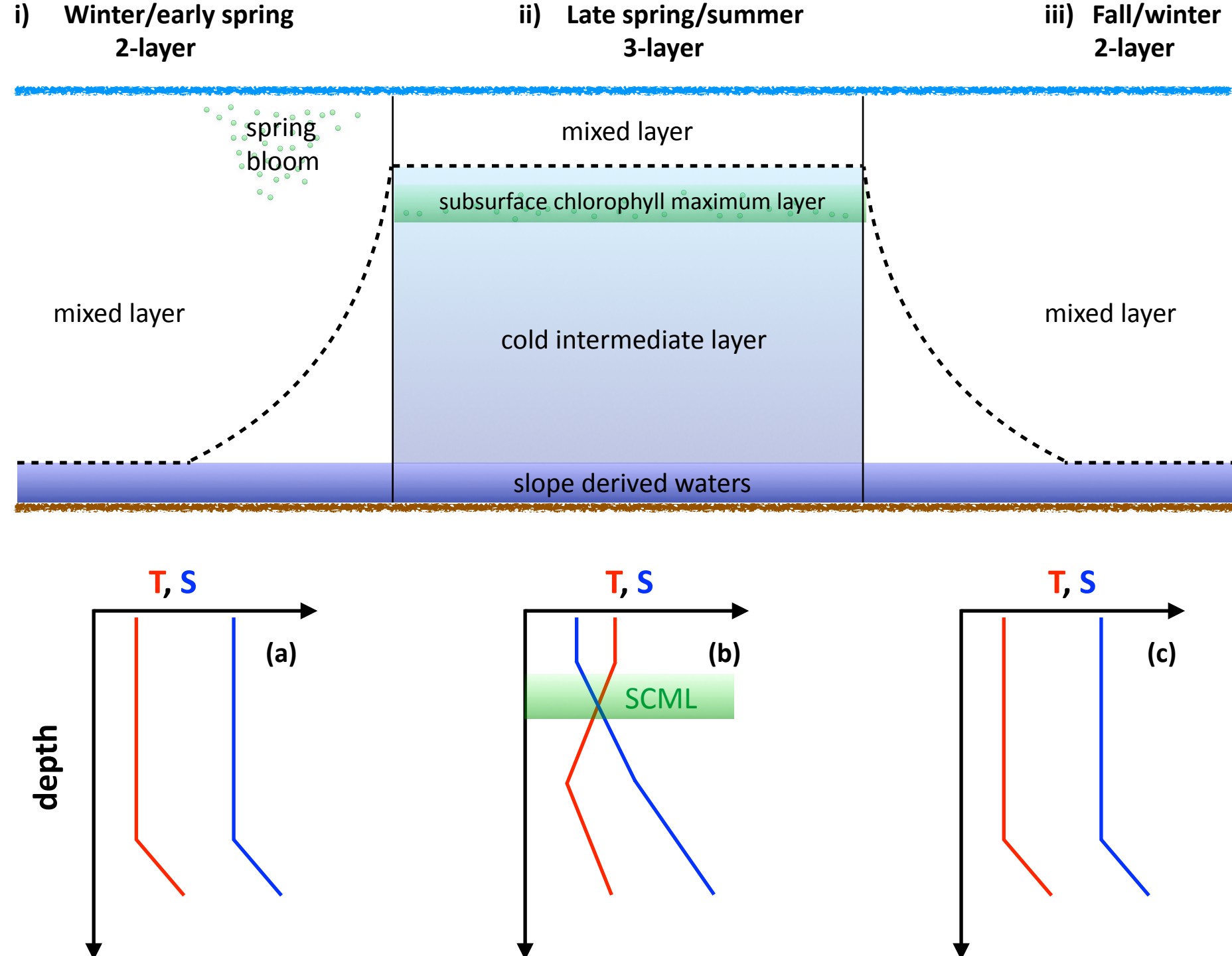

i) **Winter/early spring** **2-layer**

ii) **Late spring/summer** **3-layer**

iii) **Fall/winter** **2-layer**

spring bloom

mixed layer

subsurface chlorophyll maximum layer

mixed layer

cold intermediate layer

mixed layer

slope derived waters

T, S

T, S

T, S

(a)

(b)

(c)

SCML

depth

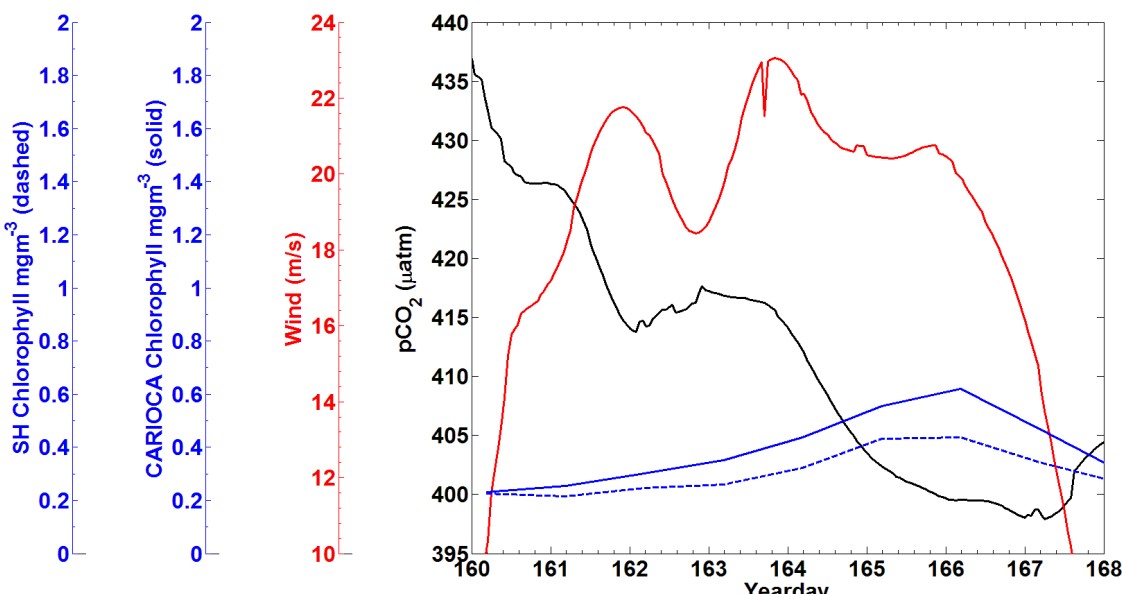

Lemay et al., Figure 4

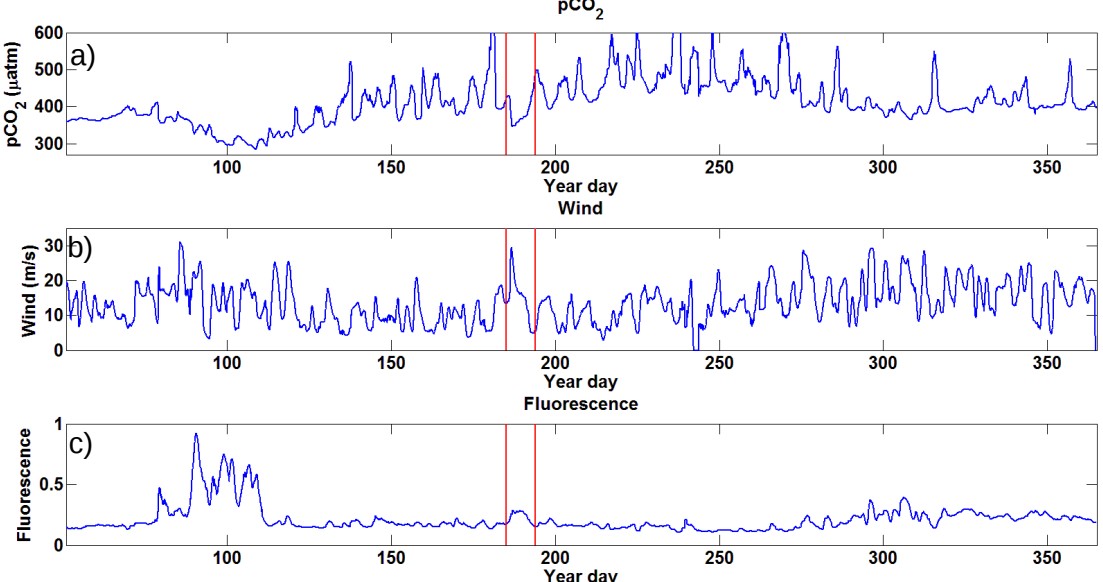

Lemay et al., Figure 5

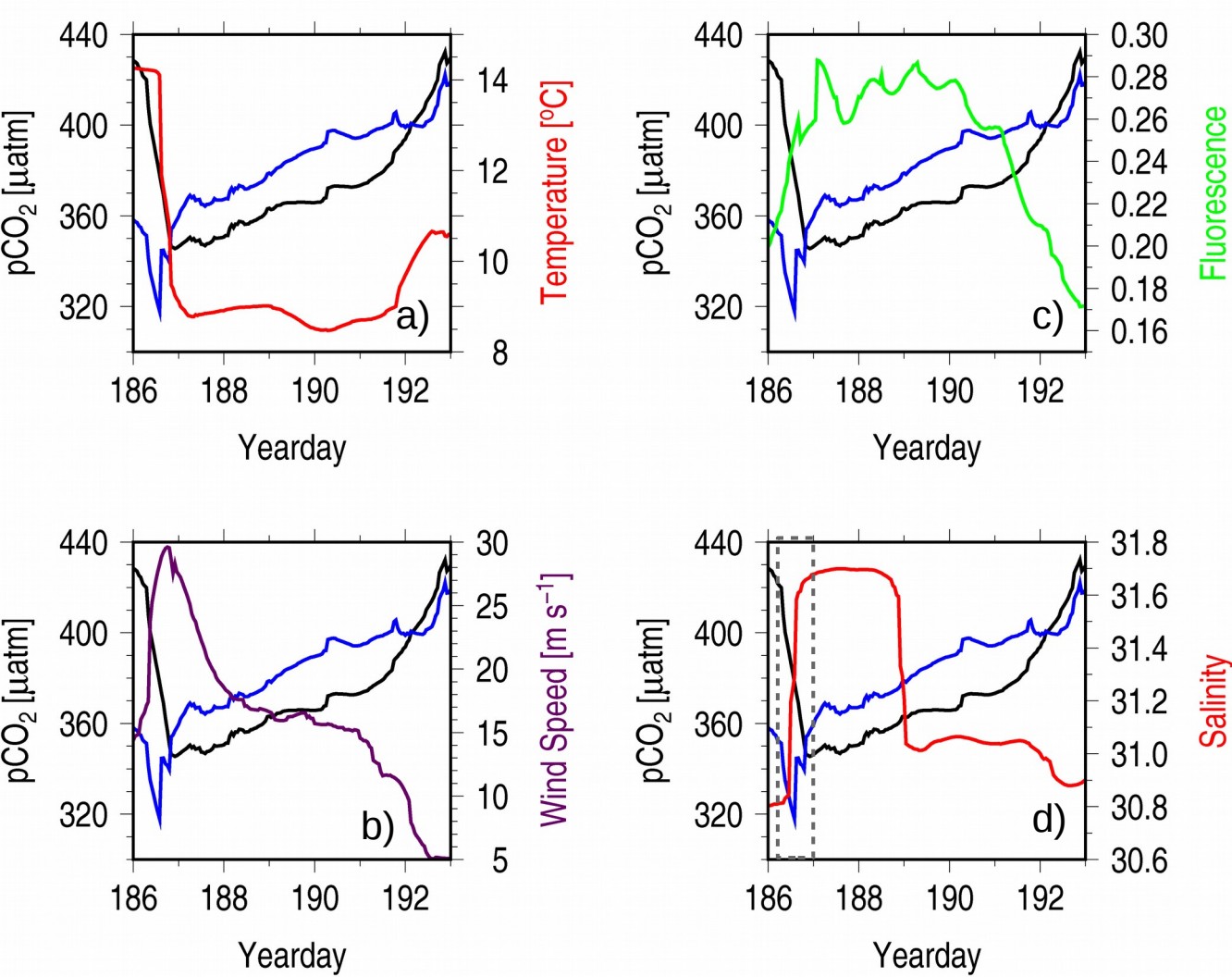

Lemay et al., Fig. 6

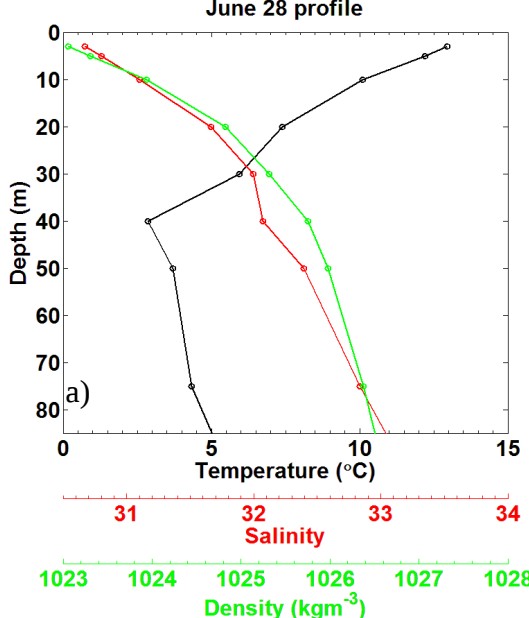

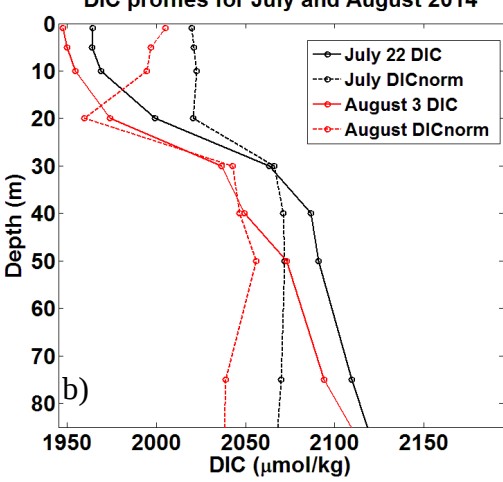

Lemay et al., Figure 7

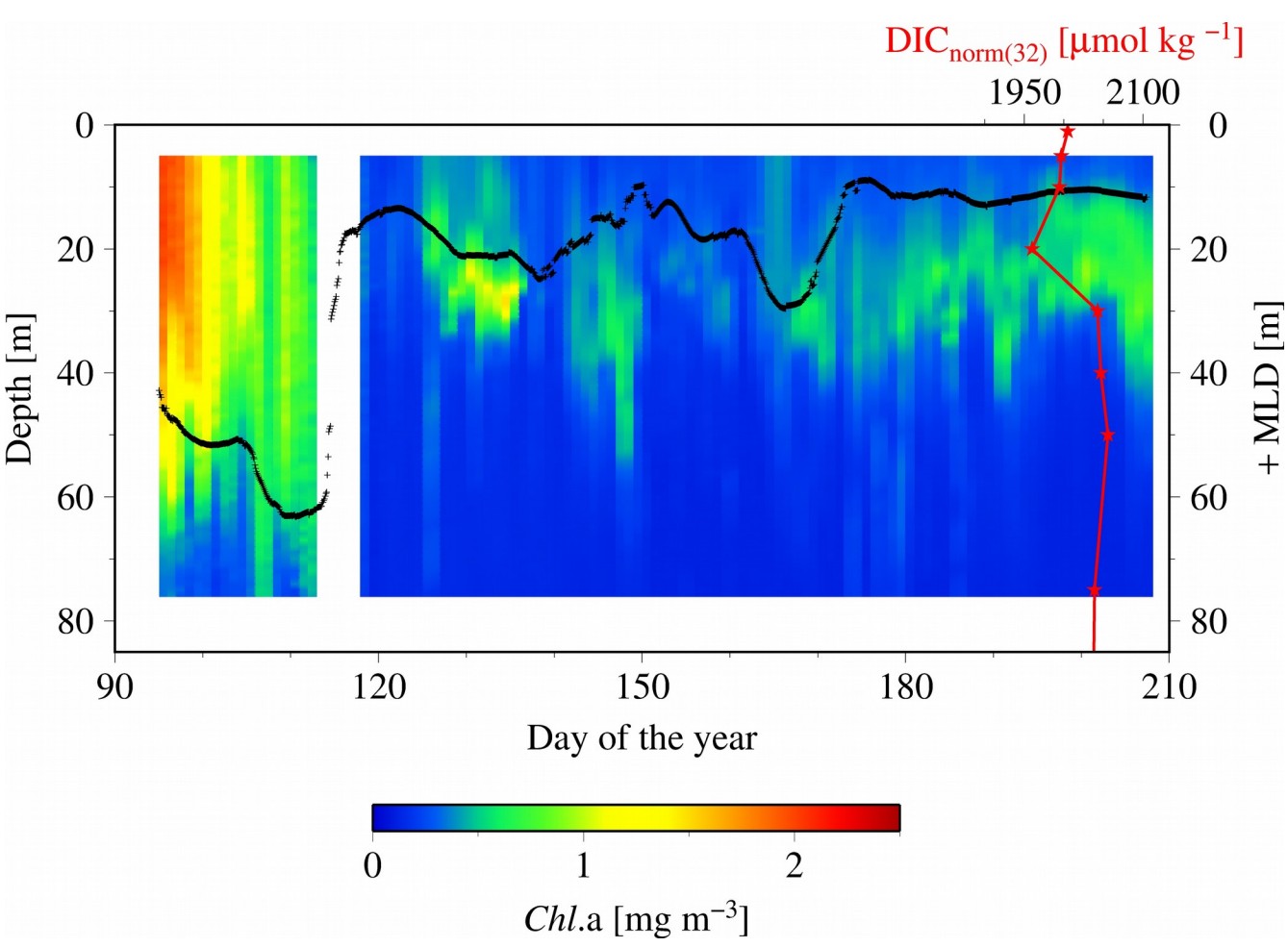

Lemay et al., Figure 8

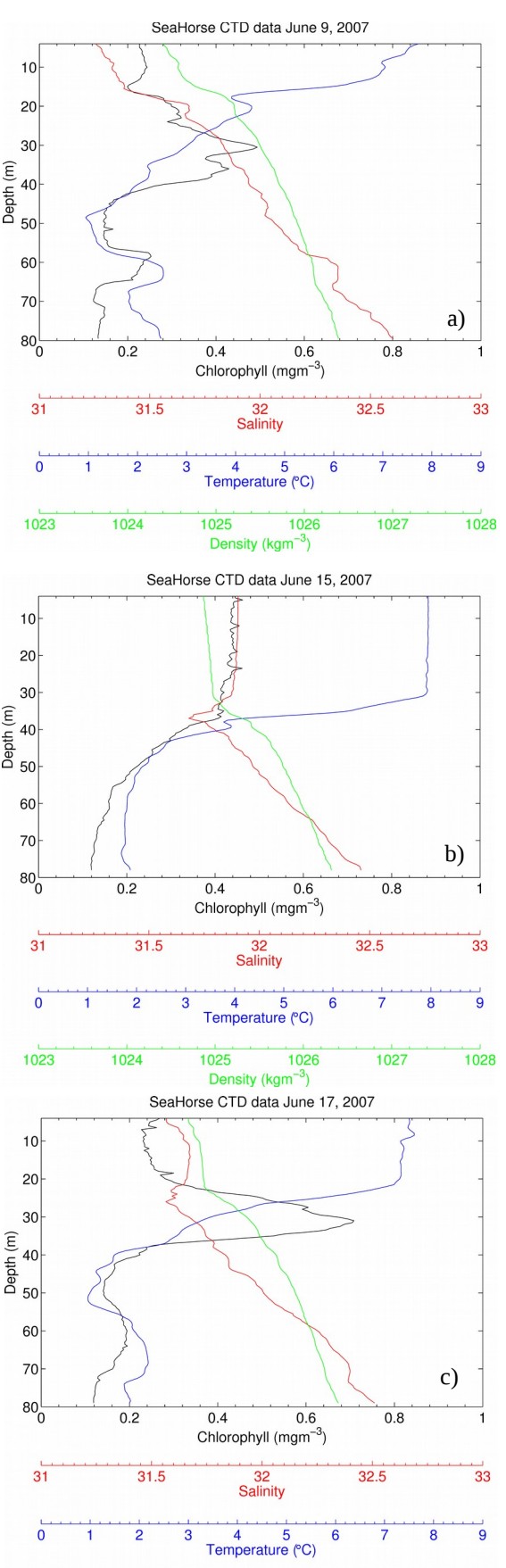

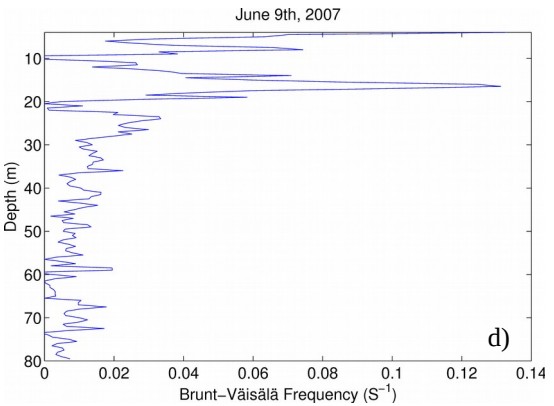

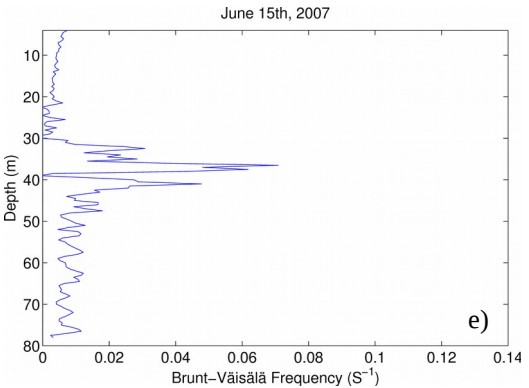

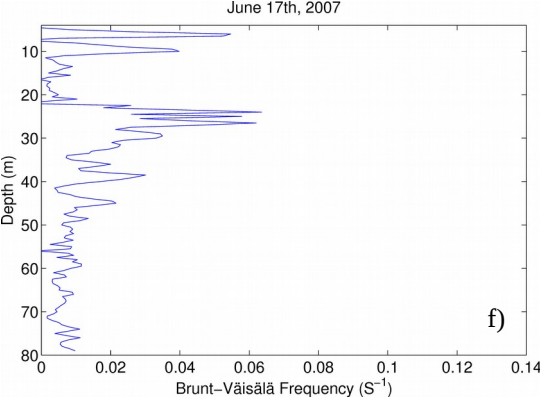

Lemay et al., Fig. 9