# Peer review of "Hurricane Arthur and its effect on the short-term variability of $pCO_2$ on the Scotian Shelf, NW Atlantic"

_Biogeosciences, 2017_

## Referee Comment (RC1) · Anonymous Referee #1 · 20 Oct 2017

**Overall Statements:**

The manuscript "Hurricane Arthur and its effect on the short-term variability of pCO2 on the Scotian Shelf, NW Atlantic" by J. Lemay, H. Thomas, S.E. Craig, W.J. Burt, K. Fennel, and B.J.W. Greenan presents the interaction between physical and biogeochemical processes on the Scotian Shelf, an open shelf sea with a complex water mass structure. The manuscript focusses on a strong wind event in July 2014. As shown, similar events emerge in this area very often. The authors did the first step in broadening the studied time interval by applying a spectral analysis. But this analysis has no further consequences within the manuscript. I would suggest to omit the spectral

analysis or to use its results for further storm event-related carbon flux estimates on longer time scales (or other biogeochemical analysis).

The manuscript is well structured and is equipped with mostly significant figures, but it contains several partly severe errors. The conclusion repeats more or less the findings. This section could be used for more general statements on storm effects on biogeochemical fluxes.

Detailed remarks:

L29: land, ocean, sediment, and atmosphere

L60: give the extent of the Scotian Shelf (lon1-lon2, lat1- lat2)

L61: at which position are the annual cycles in Fig. 2 valid?

L61ff: Which is the origin of the deep high salinity water?

L70: Indicate "CIL" in Fig. 3

L73: I do not see 20 °C in Fig. 3i

L75: The given salinity range does not fit to Fig. 2ii

L85: Fig. 2iv

L131: At which depth are the measurements taken?

L131: Give here the time interval when the buoy was applied.

L175: You mean DICS? Where S is upper case?

L176: represents the freshwater end member

L178: + DICS=0

L182: In Wanninkhof (2014) the gas transfer velocity has the unit (cm/h), so I would expect another constant to end up with mol m-2 s-1.

L205: Give a motivation for the choice of the time interval.

L234: The denoted time interval in Fig. 5 does not fit to the time interval in the text.

L263: "Figure 8 also shows that the density steadily increases with depth (Fig. 8a), and .."

L264: You combine T/S profiles from June 28 (other year?) with DIC profiles in July/August 2014. Why is this valid?

L262: My mixing calculations result in a depth of 40-50m. Mixed T=9  $^{\circ}$ C. Upper value 14 $^{\circ}$ C. Makes 4 $^{\circ}$ C as lower value, to be found at 50m depth.

L278: This sentence fits to my calculation (40-50m).

L301: There must be other sources of heat. Mixing alone should have reduced surface temperature. Please discuss this.

L301: From day 186 to the maximum value I see an increase of 40%.

L335: For a reduction of one unit in salinity the mixing should have taken place from the surface to about 70m depth (compare Fig. 2)

L342: where does this number (45 mmolC m-2 day-1) come from? Which C:Chl ratio did you use?

L376: Reference missing.

L495: Give position.

L519: Why "Climatologies"?

Fig. 5: Please give more time ticks.

L527: which DIC profile is used? July 22 or Aug 3?

L530: refer to Fig. 8

Fig. 11: Omit this figure. It is not necessary. Omit also "Figure 11" in L365.

---

## Referee Comment (RC2) · Anonymous Referee #2 · 9 Nov 2017

The authors present high-resolution biogeochemical data from the Scotian Shelf (Northwestern Atlantic) before, during, and after a hurricane event. Hourly pCO2 data re used to assess the short-term impact of the storm on the surface and subsurface properties of the water column, and the resulting impact on the air-sea CO2 exchange. The paper reports that there is a layer of cold water depleted in inorganic carbon (DIC) just above the thermocline, which is attributed to a population of phytoplankton that grows under reduced light conditions, assuming sufficient nutrients are available. The presence of the phytoplankton is confirmed by chlorophyll data, which the authors treat qualitatively having shown some disagreement between measured and sensor observations of fluorescence. With a storm event, the layer of high-biomass (and reduced

DIC) is entrained into the surface layer and results in an undersaturation in pCO2 that drives a flux of CO2 from the atmosphere to the ocean (uptake). This short-term event is found to be comparable to the spring bloom in terms of contribution to the uptake of CO2, and thus short-term wind events may have a large impact on the annual CO2 exchange in this region.

The paper is well written and structured, and most assumptions are satisfactorily justified. I believe the paper is suitable for publication in Biogeosciences following some minor revisions based on my comments below:

Line 151: was the pCO2 really measured using the VINDTA 3C – I was not aware this was possible. I thought pCO2 was computed on the basis of the DIC and TA analyses? Please clarify.

Line 288: While I understand that the SeaHorse profile data was not available for the hurricane observations, it does seem odd to rely so heavily on subsurface data from a short period several years earlier. I think the text would benefit from more information/validation about these data and how representative they are of the conditions preceding the 2014 storm event. Is there other climatological data that could be used to provide greater context for these short term observations below the surface?

Conclusions: I found the description of the schematic to be oddly placed in the conclusion – please consider relocating to the discussion. I also found Figure 11 to be a somewhat confusing representation of the more clearly described mechanistic understanding of the system in the text.

I believe the other reviewer suggested that Fig. 11 was not necessary, and I'm inclined to agree. If you do want to include a schematic, I would suggest coming up with something that has multiple panels contrasting the situation where there is a short-term wind event with when there isn't – or a time evolution of the 2 to 3 layer system. As shown it does not convey the arguments the author's are trying to make.

Caption for Fig. 7: I don't see how these are "climatologies"?

---

## Author Comment (AC1) · 15 Dec 2017

Response to Referee 1:

We very much appreciate this detailed and insight full review. It has helped substantially improve the manuscript. We respond to the referee's points below, and have adopted the points for any revised version of the manuscript accordingly.

"Overall Statements: The manuscript "Hurricane Arthur and its effect on the shortterm variability of pCO2 on the Scotian Shelf, NW Atlantic" by J. Lemay, H. Thomas, S.E. Craig, W.J. Burt, K. Fennel, and B.J.W. Greenan presents the interaction between

physical and biogeochemical processes on the Scotian Shelf, an open shelf sea with a complex water mass structure. The manuscript focuses on a strong wind event in July 2014. As shown, similar events emerge in this area very often. The authors did the first step in broadening the studied time interval by applying a spectral analysis. But this analysis has no further consequences within the manuscript. I would suggest to omit the spectral analysis or to use its results for further storm event-related carbon flux estimates on longer time scales (or other biogeochemical analysis). "

We agree to omit the spectral analysis, as it, in essence, confirms results of earlier studies. We now reworded the section stating that the data used in the present paper reflect the (reoccurring) winter storm pattern as reported by Smith et al., 1987, Shadwick et al., 2010, or Thomas et al., 2012, but do not present our own analysis and Fig. 6 any longer.

"The manuscript is well structured and is equipped with mostly significant figures, but it contains several partly severe errors. The conclusion repeats more or less the findings. This section could be used for more general statements on storm effects on biogeochemical fluxes. "

We have thoroughly checked the manuscript for severe and less severe errors and apologize for having overlooked these. In the concluding section, we have placed storm events in a broader perspective.

Detailed remarks: L29: land, ocean, sediment, and atmosphere We modified the text accordingly.

L60: give the extent of the Scotian Shelf (lon1-lon2, lat1- lat2) This information has been added to the text (43N-46N, 66W-60W).

L61: at which position are the annual cycles in Fig. 2 valid? This information has been added to the caption of Fig. 2.

L61ff: Which is the origin of the deep high salinity water? This information has been
added to the caption to the section (the warm slope water).

L70: Indicate "CIL" in Fig. 3 The CIL has already been indicated in our original figure, spelled out though.

L73: I do not see 20  $\hat{a}UqC$  in Fig. 3i We agree with this point, however think that this is an issue between observed peak values (20C) and long-term mean values (15C). We have reworded the section to solve this.

L75: The given salinity range does not fit to Fig. 2ii Thank you, this was a typo on our side.

L85: Fig. 2iv Thank you, this was a typo on our side.

L131: At which depth are the measurements taken? Measurements were taken at the surface at approx. 1m depth. We have added this information to the document.

L131: Give here the time interval when the buoy was applied. From February 20th to December 31st We have added this information to the document.

L175: You mean DICS? Where S is upper case? Thank you, this was a typo on our side.

L176: represents the freshwater end member We modified the text accordingly.

L178: + DICS=0 Thank you, this was a typo on our side.

L182: In Wanninkhof (2014) the gas transfer velocity has the unit (cm/h), so I would expect another constant to end up with mol m-2 s-1. Indeed, the unit in our text contained an error. The corrected unit is 10-5 mol m-2 hr-1.

L205: Give a motivation for the choice of the time interval. The reason for the choice of time interval is that a small storm event during that period happened while both the SEAHORSE profiler and CARIOCA buoy were in the water. We have added this explanation to the paper.
L234: The denoted time interval in Fig. 5 does not fit to the time interval in the text. Thank you, this was a typo in the text.

L263: "Figure 8 also shows that the density steadily increases with depth (Fig. 8a), and .." We modified the text accordingly.

L264: You combine T/S profiles from June 28 (other year?) with DIC profiles in July/August 2014. Why is this valid?

The profiles used in Fig. 8 are from the same year. Fig 8a shows the pre-storm conditions, Fig 8b shows the reestablishment of the system after the storm had passed. We have clarified this in the caption, thank you.

L262: My mixing calculations result in a depth of 40-50m. Mixed T=9 âŮęC. Upper value 14âŮęC. Makes 4âŮęC as lower value, to be found at 50m depth.

Thank you, this was a typo on our side.

L278: This sentence fits to my calculation (40-50m). Indeed!

L301: There must be other sources of heat. Mixing alone should have reduced surface temperature. Please discuss this. We agree with this point. We have added the following statement to the text: "As the data presented in Fig. 10 - in particular temperature - show, lateral processes, either cross-shelf or along-shelf may have impacted the system, as well. These features, however, cannot be further resolved referring to single-point moored observations."

L301: From day 186 to the maximum value I see an increase of 40%. Thank you. We assume that this comment refers to the statement in line 306. We agree the increase is approximately 40%. The text has been modified accordingly.

L335: For a reduction of one unit in salinity the mixing should have taken place from the surface to about 70m depth (compare Fig. 2) Thank you for this comment. Again we think that this is an issue between single observations and climatology means as

BGD
shown in Fig 2. We have added a statement reiterating that Fig.2 uses climatological mean values.

L342: where does this number (45 mmolC m-2 day-1) come from? Which C:Chl ratio did you use? We think that this could be a misunderstanding on the referee's side. The fluxes have been computed directly from our buoy observations. We reworded the sentence for clarity.

L376: Reference missing. We have added the reference to the list, thank you.

L495: Give position. As per our response above, we have added the position of the station to the caption.

L519: Why "Climatologies"? Thank you, we agree. We have replaced here climatologies by observations.

Fig. 5: Please give more time ticks. These will be provided in the revised figure.

L527: which DIC profile is used? July 22 or Aug 3? We used the August 3rd-profile. This has been mentioned in the caption now.

L530: refer to Fig. 8 We have modified the text accordingly.

Fig. 11: Omit this figure. It is not necessary. Omit also "Figure 11" in L365. We no longer use Fig. 11 in the paper.

BGD

---

## Author Comment (AC2) · 15 Dec 2017

Anonymous Referee #2 We are grateful for this constructive review, allowing us to clearly improve the paper.

 "The authors present high-resolution bio-geochemical data from the Scotian Shelf (Northwestern Atlantic) before, during, and after a hurricane event. Hourly pCO2 data re used to assess the short-term impact of the storm on the surface and subsurface properties of the water column, and the resulting impact on the air-sea CO2 exchange. The paper reports that there is a layer of cold water depleted in inorganic carbon (DIC) just above the thermocline, which is

attributed to a population of phytoplankton that grows under reduced light conditions, assuming sufficient nutrients are available. The presence of the phytoplankton is confirmed by chlorophyll data, which the authors treat qualitatively having shown some disagreement between measured and sensor observations of fluorescence. With a storm event, the layer of high-biomass (and reduced DIC) is entrained into the surface layer and results in an undersaturation in pCO2 that drives a flux of CO2 from the atmosphere to the ocean (uptake). This short-term event is found to be comparable to the spring bloom in terms of contribution to the uptake of CO2, and thus short-term wind events may have a large impact on the annual CO2 exchange in this region.

The paper is well written and structured, and most assumptions are satisfactorily justified. I believe the paper is suitable for publication in Biogeosciences following some minor revisions based on my comments below:

Line 151: was the pCO2 really measured using the VINDTA 3C – I was not aware this was possible. I thought pCO2 was computed on the basis of the DIC and TA analyses? Please clarify.

Thank you. This was a typo on our side. DIC has been measured by the VINDTA. For clarification the CARIOCA buoy performs direct measurements of the pCO2. We do not present any pCO2 values here, which are computed from DIC and TA, only direct pCO2 observations.

Line 288: While I understand that the SeaHorse profile data was not available for the hurricane observations, it does seem odd to rely so heavily on subsurface data from a short period several years earlier. I think the text would benefit from more information/validation about these data and how representative they are of the conditions preceding the 2014 storm event. Is there other climatological data that could be used to provide greater context for these short term observations below the surface? Conclusions: I found the description of the schematic to be oddly placed in the conclusion – please consider relocating to the discussion. I also found Figure 11 to be a somewhat

confusing representation of the more clearly described mechanistic understanding of the system in the text.

In accordance with reviewer 1 we have attempted to clarify the supporting use of climatological data and data from other years throughout the paper. Further we have deleted Figure 11 from the manuscript.

I believe the other reviewer suggested that Fig. 11 was not necessary, and I'm inclined to agree. If you do want to include a schematic, I would suggest coming up with something that has multiple panels contrasting the situation where there is a short-term wind event with when there isn't – or a time evolution of the 2 to 3 layer system. As shown it does not convey the arguments the author's are trying to make.

Again, we agree with to point, have deleted the Figure 11, and think that Figure 3 shows the situation properly.

Caption for Fig. 7: I don't see how these are "climatologies"?

We have adopted this suggestion, which also has been made by referee 1.

---

## Referee Report (RR1)

The text was revised thoroughly by the authors. All my questions are answered. Unfortunately the conclusions are still a bit poor. Please give at least a reference on the assumed storm frequency increase. As far as I know this point is still under debate. The connection (after the excursion to climate change) to the next sentence ("It is anticipated ..") is not given.